# Impacts of Invasive Plants on Native Vegetation Communities in Wetland and Stream Mitigation

**DOI:** 10.3390/biology13040275

**Published:** 2024-04-18

**Authors:** Douglas A. DeBerry, Dakota M. Hunter

**Affiliations:** 1Environment and Sustainability Program, College of William & Mary, Williamsburg, VA 23187, USA; 2Biology Department, College of William & Mary, Williamsburg, VA 23187, USA; dhunter@copperheadconsulting.com

**Keywords:** wetland mitigation, stream mitigation, invasive plant species, invasive plant management, ecological performance standard

## Abstract

**Simple Summary:**

Vegetation communities are impacted by invasive plants in predictable ways, namely, reductions in species richness, diversity, and floristic quality (i.e., the “nativeness” of the plant community). In ecological restoration, these plant community properties are important in establishing performance standards, especially for restored wetlands and streams. Performance standards can be thought of as the “report card” for ecological restoration. In the United States, much of this ecosystem restoration occurs as compensatory mitigation (i.e., to compensate for impacts to wetlands and streams elsewhere). The presence of invasive plant species is an important performance standard used in compensatory mitigation to determine whether or not wetland and stream mitigation sites comply with environmental laws. Invasive plants detract from restoration performance, but it is unclear what level of invasion should trigger a legal requirement for invasive plant removal. This study found that lower levels of invasion (i.e., 5–10%) do not diminish native vegetation community properties on wetland and stream mitigation sites, so low invasive performance standards could be causing more harm than good via the loss of native species from treatments like broad-spectrum chemical herbicides. Our research points to a more moderate standard of 10%, along with annual invasive mapping.

**Abstract:**

We sampled vegetation communities across plant invasion gradients at multiple wetland and stream mitigation sites in the Coastal Plain and Piedmont physiographic provinces of Virginia, USA. Impacts of invasion were evaluated by tracking changes in species composition and native vegetation community properties along the abundance gradients of multiple plant invaders. We found that native species richness, diversity, and floristic quality were consistently highest at moderate levels of invasion (ca. 5–10% relative abundance of invader), regardless of the identity of the invasive species or the type of mitigation (wetland or stream). Likewise, native species composition was similar between uninvaded and moderately invaded areas, and only diminished when invaders were present at higher abundance values. Currently, low thresholds for invasive species performance standards (e.g., below 5% relative abundance of invader) compel mitigation managers to use non-selective control methods such as herbicides to reduce invasive plant cover. Our results suggest that this could cause indiscriminate mortality of desirable native species at much higher levels of richness, diversity, and floristic quality than previously thought. From our data, we recommend an invasive species performance standard of 10% relative invader(s) abundance on wetland and stream mitigation sites, in combination with vigilant invasive plant mapping strategies. Based on our results, this slightly higher standard would strike a balance between proactive management and unnecessary loss of plant community functions at the hands of compulsory invasive species management.

## 1. Introduction

One of the most important and pervasive contemporary issues in the field of ecological restoration is that of biological invasion [1]. Invasive species are generally characterized by explosive population growth in combination with a highly competitive life history strategy [2], and this is especially problematic in restoration because invaders can quickly pre-empt space that might otherwise be occupied by desirable species [3]. Disturbance represents a mode of introduction for invasive plants, and ecological restoration sites can be particularly susceptible to biological invasion because the practices used to create, restore, or enhance ecological conditions are often the same types of disturbances that leave a site vulnerable to invasion (e.g., site clearing and grading, etc. [4,5]).

When ecological restoration is undertaken to compensate for impacts to ecosystems elsewhere, it is generally referred to as “compensatory mitigation.” In the United States, a large percentage of compensatory mitigation is completed under the purview of Section 404 of the Clean Water Act (USC 33 §1344 et seq.) and analogous state water control law and includes wetlands or streams that are created or restored to compensate for impacts to similar ecosystems permitted through federal and/or state regulatory programs [6]. In the Mid-Atlantic region, the largest percentage of compensatory wetland and stream mitigation is non-tidal [7], which is the focus of our research. Hereafter, “non-tidal compensatory wetland and stream mitigation” will be referred to collectively as “compensatory mitigation” or simply “mitigation”. When addressed separately, the terms “wetland mitigation” and “stream mitigation” will be used.

On compensatory mitigation sites in the Mid-Atlantic region, invasive plant species present one of the greatest challenges to mitigation managers, ecological restoration designers, and natural resource agencies [8]. The expense of managing plant invaders has increased considerably over the past couple of decades, and in some cases, it can represent the largest investment of money and resources on mitigation sites [9]. This is happening without a clear understanding of secondary effects from using aggressive management techniques like chemical herbicides to kill invasive plants.

One reason for this is that biological invasion is a relatively new subject of study to science, deriving many of its first principles from agriculture or other commodity-based disciplines (e.g., mariculture, silviculture, etc.) [10]. In these fields of research, the emphasis has been on management programs that will maximize values (i.e., ecosystem attributes beneficial to mankind), with less emphasis on maximizing ecological functions (i.e., ecosystem attributes beneficial to the ecosystems themselves). Although some research has addressed invasion and function on mitigation sites (e.g., [11,12,13,14]), in most cases, invasive species have been ancillary to the primary research objectives.

### 1.1. Invasive Species Performance Standards in Compensatory Mitigation

Perhaps even more important is the issue of performance standards for invasive species in compensatory mitigation. Performance standards are established to ensure that aquatic resource functions are maximized on mitigation sites, but it is unclear how invasive species standards accommodate this goal. For example, a standard that is set low (e.g., a 5% threshold for invasive plant cover, which was the de facto performance standard in Virginia at the time of this study [15]) often necessitates the use of targeted or broadcast herbicides, a practice that introduces foreign chemicals into natural systems and can result in collateral damage to desirable species [16,17,18].

A review of federal and state mitigation policy across the U.S. shows that established requirements vary from state to state and sometimes even within individual regulatory programs. For example, Reiss et al. [19] reported a range of performance thresholds for invasive species from as low as 1% to as high as 10% in Florida, and Kozich and Halvorsen [20] and WSDE [21] documented 10% thresholds for Michigan and Washington state, respectively. In past guidance documents, Ohio set a 5% threshold for non-*Typha* invaders, but up to 10% for *Typha* spp. due to challenges in differentiating native species from hybrids in that genus [22]. In some project-specific instances, Ohio mitigation banks have been established with a non-specific performance standard requiring the overall vegetative community to be “predominantly native” [23]. As noted above, similar qualitative criteria have been specified for mitigation projects in Illinois, where the invasive or nuisance species standard was “none dominant” [12]. Maryland adopted a similar standard with a bit more specificity in requiring that mitigation sites could not be “dominated by common reed (*Phragmites australis*) or other nuisance vegetation”, a standard aimed at one of the more problematic invaders in that state [24]. In their Mitigation Banking Instrument (MBI) Template, North Carolina simply stated that invasive species could not impact the “functional integrity of the target vegetative community”, but functional integrity was not clearly defined in the standard [25]. As this is just a selection of the many regulatory programs and mitigation guidance documents in the U.S., it is evident that invasive species performance in compensatory mitigation lacks consensus.

### 1.2. Mitigation Plant Communities and Invasive Species

One point of widespread agreement in the literature is that the presence of invasive plants on mitigation sites has a diminishing effect on ecosystem function [8,11,12,26,27]. Most of these studies evaluated ecosystem effects after the invaders had become well established in the community, but what is of interest to mitigation practitioners is the ecological effects of invaders in the *early* stages of invasion. This is because regulatory agencies are requiring managers to control invaders at these lower abundance values.

Although the literature is generally lacking on the topic “early mitigation invasion,” some studies on vegetation ecology in non-tidal wetland mitigation have demonstrated relevant trends. For example, Perry et al. [28] summarized cattail (*Typha* spp.) studies on mitigation sites, concluding that the standard rationale for cattail removal—namely, that cattails reduce species richness and diversity within the vegetative community—is not supported by the research. Further, although research reported by DeBerry and Perry [29] did not focus specifically on invasive species, datasets from this study of fifteen mitigation wetlands showed that sites where certain invasive species were dominant (e.g., *Typha latifolia*, *Microstegium vimineum*, and *Lespedeza cuneata*) also had among the highest species richness values. Interpreting similar data from [27], species richness and diversity index values for mitigation sites with invasive species (e.g., *M. vimineum* and *Murdannia keisak*) were not statistically different from the same indices calculated for sites with no invasives.

The purpose of this research was to address the above considerations by answering the following questions: (1) How do invasive plant species impact ecosystem functions related to native plant composition, richness, floristic quality, and diversity on compensatory mitigation sites? (2) Are current invasive species performance standards in mitigation aligned with #1 above and, if not, are there other standards that are more congruent with the magnitude of the problem? We accomplished these objectives by measuring vegetation community properties across invasion gradients on multiple wetland and stream mitigation sites throughout the Coastal Plain and Piedmont physiographic provinces in Virginia. On our research sites, the “invasion gradient” was represented by the transition from high to low abundance of a target invader, which was evaluated in this study using plots arrayed on transects across the gradient.

## 2. Materials and Methods

### 2.1. Target Invaders

We met with several mitigation professionals (e.g., environmental consultants, mitigation bankers, agency representatives, etc.) to develop a list of the most problematic invasive plants on mitigation sites in the region. From this list, we selected five target invaders based on site-level criteria (see Section 2.2 below). Two were more problematic on wetland mitigation sites (*Arthraxon hispidus*, *Typha* spp.), two were more common on stream mitigation sites (*Lespedeza cuneata*, *Lonicera japonica*), and one was prevalent on both wetland and stream sites and was therefore included in both datasets (*Microstegium vimineum*). A summary of target invaders is provided in Table 1. For brevity, invaders will be referred to by genus names hereafter (i.e., *Arthraxon*, *Lespedeza*, *Lonicera*, *Microstegium*, and *Typha*).

### 2.2. Study Sites

Field sites were selected based on the following suitability criteria: (1) they had to be established as non-tidal forested wetland or stream mitigation under the regulatory purview of federal and state environmental laws, and (2) they had to have dominant populations of the invasive plants from our target list of invaders. We acquired planning documents for all sites from the owners or mitigation professionals who designed them and determined that all sites were generally constructed in a similar manner; i.e., all required some amount of earth moving to establish final elevation grades, and all were planted with trees and shrubs as well as a native herbaceous seed mix. As the sites were situated in the Piedmont and Coastal Plain physiographic provinces in Virginia; they were generally in similar landscape positions and had similar geomorphic features (wetland and stream sites, respectively).

The wetland mitigation study was completed over the 2017 and 2018 growing seasons. Out of 30 mitigation sites evaluated for inclusion, 23 met suitability criteria and were selected for sampling. Wetland sites ranged in age from 1 to 23 years post-construction (i.e., following final site development and planting) and were evenly distributed across the Piedmont (11 sites) and Coastal Plain (12 sites) in Virginia (Figure 1).

The stream mitigation field study was completed over the 2018 and 2019 growing seasons, with 21 sites chosen for sampling from 30 initially screened. Stream sites ranged in age from 1 to 19 years post-construction and were also evenly distributed across the Piedmont (10 sites) and Coastal Plain (11 sites) in Virginia (Figure 1).

Although site age has been implicated as an important factor in vegetation development on mitigation sites [29,61], it was not explicitly included as a factor in our analysis because we were focused on sites with existing invasion gradients irrespective of site age. However, during the site selection process, we ensured that all datasets had an even distribution of sites from five age classes: 1–2 years, 3–5 years, 6–10 years, 11–15 years, and >15 years post-restoration. Suffice it to mention that site age was evaluated in a related study on environmental drivers of invasion in mitigation and was not a significant site-level factor for most of the target invaders in that analysis [62].

### 2.3. Vegetation Sampling

We established the invasion gradient on all sites by conducting an initial site screening to locate representative populations of the target invaders. For vegetation sampling, we selected invasive populations where the apparent change in environmental conditions was negligible from the invaded end of the gradient to the uninvaded end (e.g., same relative elevations, same apparent hydrology regime, etc.).

#### 2.3.1. Wetland Mitigation Sampling Methods

Within representative populations of each target invader on wetland sites, linear transects were established across the invasion gradient from “completely invaded” (i.e., dominant, or greater than 20% relative cover) to “uninvaded” (i.e., less than 5% relative cover). Five plots were arranged along each transect using a randomization procedure to determine plot centers and transect direction (Figure 2a). Plot A corresponded to “completely invaded”, Plot C approximated the “edge” of the invasive population, and Plot E was at the “uninvaded” end of the transect. Plots B and D were established in sequence. The sample area at each plot was 4 m^2^ and comprised of four 1 m^2^ nested sampling frames arranged in the four quadrants surrounding the plot center (vertex).

We quantified vegetation abundance using cover estimates for all species within each of the four 1m^2^ subplots nested in the 4m^2^ plots. Cover estimates were based on a modified Daubenmire cover class scale with midpoints used for analysis [63]. The cover classes, with midpoints in parentheses (rounded to the nearest whole integer), included the following: 0–1% (1%); 1–5% (3%); 5–25% (15%); 25–50% (38%); 50–75% (63%); 75–95% (85%); and 95–100% (98%). Cover classes were recorded for each species and then averaged across the four 1m^2^ subplots. Identifications of all vascular plants were either obtained onsite or samples were gathered and preserved for later verification. Intact collections were deposited at the College of William & Mary Herbarium (WILLI) following confirmation of identity by a senior botanist. Nomenclature follows Weakley et al. [64]. Native/non-native status was based on Virginia Botanical Associates [65] and Weakley et al. [64].

#### 2.3.2. Stream Mitigation Sampling Methods

For the stream sites, sampling design and approach followed the wetland methods outlined above with one exception: instead of using a randomly defined direction to establish a straight line transect, plots were randomized at each location along a transect that meandered roughly parallel to the nearest streambank to maintain a consistent relative elevation in the floodplain (Figure 2b). The purpose for this modification was to ensure that landscape position within the floodplain was similar for each plot along the invasion gradient. Plot dimensions and cover estimation techniques were the same as described above.

### 2.4. Data Synthesis and Analysis

Data analysis was completed using R version 4.0.3 [66] including the packages vegan, Hmisc, and BiodiversityR [67,68,69,70]. The datasets for each invasive species were analyzed separately due to expected variation in their relative tolerances for environmental stressors and discrepancies among growth requirements [3,30]. Across the invasion gradient, changes in species composition were assessed with the Sørensen similarity index [63], the significance of which was tested via analysis of similarity (ANOSIM) [67]. Floristic quality index (FQI) was calculated based on DeBerry and Perry [71] using the most recent coefficients of conservatism (C-values) for the Virginia flora [72]. Community properties were evaluated with species accumulation curves (species richness) and Rényi profiles (species diversity) [67]. All permutation tests of significance were set at 1000 iterations, and statistical analyses were evaluated at α = 0.05.

### 2.5. Analysis for Ecological Performance Standard

To determine invasion thresholds for establishing reasonable ecological performance standards, we sorted each community matrix in descending order of invasive species dominance and plotted the running average of relative invasive species abundance against the running average of native species richness. Native richness was chosen because of its importance in vegetation performance standards for compensatory mitigation [12,71,73], and also because native species trends were representative of the other floristic quality parameters evaluated in our results for every dataset (namely, composition, FQI, evenness, and diversity). The “bin” size for each average calculation was equivalent to the original bin size, or total number of transects, for each group (e.g., 14 for *Typha*, 10 for *Arthraxon*, etc.). Calculated in this way, we were able to superimpose the trend in native richness over the invasion gradient and observe the point at which the “hump” in native richness began to decline on the invaded side of the gradient, which was visualized by fitting a polynomial trendline to the scatterplot of native richness data points.

## 3. Results

### 3.1. Wetland Study

One hundred ninety-four (194) species were documented in the overall wetland mitigation study across 23 sites, 34 transects, and 170 plots sampled. A checklist of species encountered is included in Appendix A. Community data are summarized below for each of the three target invasive species.

### 3.2. Species Composition—Wetlands

*Arthraxon*: In the *Arthraxon* community dataset, 124 species were sampled from 50 plots along 10 transects. *Arthraxon* comprised 19.5% of the overall relative abundance within the community matrix. Co-dominants (calculated using the 50/20 rule [74]) included *Leersia oryzoides* (8.2%), *Symphyotrichum racemosum* (6.1%), *Juncus effusus* (5.2%), *Salix nigra* (3.8%), *Fraxinus pennsylvanica* (3.6%), *Platanus occidentalis* (2.7%), and *Eleocharis tenuis* (2.6%). The Sørensen similarity matrix for the *Arthraxon* dataset showed that community composition was somewhat similar across the invasion gradient (Table 2(a)), with all values close to a similarity cutoff of 0.5 for the index as defined by Mueller-Dombois and Ellenberg [63]. Analysis of similarity (ANOSIM) suggested a significant between-group difference based on permutations (*p* = 0.003). From inspection of the ANOSIM boxplots (Figure 3a), nearly all between-group variation was attributable to the A (most invaded) group, but the B (second most invaded), C (moderately invaded), D (second least invaded), and E (uninvaded) groups were strongly aligned with between-group similarity and therefore compositionally similar.

*Microstegium* (wetlands): The *Microstegium* wetland community dataset included 116 species sampled from 50 plots across 10 transects. *Microstegium* comprised 20.6% of the overall relative abundance within the community matrix. Co-dominants included *Acer saccharinum* (7.7%), *Scirpus cyperinus* (5.9%), *Fraxinus pennsylvanica* (5.7%), *Pinus taeda* (5.4%), *Betula nigra* (4.4%), and *Juncus effusus* (3.7%). As with the *Arthraxon* analysis, the *Microstegium* Sørensen matrix showed marginal compositional similarity across the invasion gradient (Table 2(b)). ANOSIM results demonstrated a significant between-group difference (*p* = 0.001), and boxplots indicated that this difference was due to the invaded groups (A and B), with C, D, and E groups being compositionally similar (Figure 3b).

*Typha*: The *Typha* community matrix included 106 species sampled from 70 plots across 14 transects. *Typha* accounted for 19.5% of the overall relative abundance, with co-dominants *Persicaria hydropiperoides* (11.6%), *Juncus effusus* (10.8%), *Leersia oryzoides* (7.7%), and *Scirpus cyperinus* (4.9%). As above, the *Typha* community matrix showed marginal similarity in species composition across the invasion gradient based on Sørensen index values (Table 2(c)). ANOSIM results showed significant between-group variation (*p* = 0.003), and boxplots indicated that this was attributable to the most invaded group (A), with the remaining groups showing overlap and compositional similarity (Figure 3c).

### 3.3. Community Properties—Wetlands

*Arthraxon*: In the *Arthraxon* community matrix, native species richness peaked at moderate levels of invasion (group C) and no invasion (group E) across the gradient, and FQI was highest at moderate levels of invasion (Table 3). These results accord with species accumulation curves and Rényi diversity profiles, which showed moderately invaded plots (group C) among the highest in species richness (Figure 4a), and consistently highest in diversity and evenness (Figure 4d). It is important to note that in the case of the *Arthraxon* dataset, species richness by itself provided only marginal differentiation among groups along the invasion gradient from A (most invaded) to E (uninvaded), as the accumulation curves for most groups were close and somewhat overlapping (Figure 4a). However, the Rényi diversity profiles, which account for species richness, evenness, and diversity, indicate that moderate levels of invasion (C) correspond to the highest values of these community metrics (Figure 4d). All results in the *Arthraxon* dataset confirmed that the highest levels of *Arthraxon* invasion (group A) negatively affected species richness, diversity, and evenness.

*Microstegium* (wetlands): Similar to *Arthraxon*, native species richness and FQI were highest at moderate levels of invasion for *Microstegium* (Table 3). Likewise, species accumulation curves showed a clear pattern of species richness values where moderately invaded plots (group C) corresponded to the highest levels of richness across the dataset (Figure 4b). Rényi diversity profiles suggested similar results, although group C diversity values overlapped with group D (second least invaded) and group E (uninvaded) values (Figure 4e). These results also confirmed that the highest levels of *Microstegium* invasion (group A) negatively affected community properties.

*Typha*: As above, the *Typha* community matrix showed the highest native species richness and FQI values at moderate levels of invasion (Table 3). Species accumulation curves and Rényi profiles for the *Typha* dataset accorded with these results, showing that the moderately invaded group (C) was clearly differentiated as the most species-rich and most diverse along the invasion gradient (Figure 4c,f). As with *Arthraxon* and *Microstegium*, the highest levels of *Typha* invasion (group A) corresponded to the lowest levels of these community metrics.

### 3.4. Stream Study

Two hundred eighty-six (286) species were documented in the overall stream mitigation field study across 21 sites, 29 transects, and 145 plots sampled. A checklist of species encountered is included in Appendix A. Community and environmental data are summarized below for each of the three target invasive species.

### 3.5. Species Composition—Streams

*Lespedeza*: In the *Lespedeza* community dataset, 148 species were sampled from 40 plots along eight transects. *Lespedeza* comprised 17.5% of the overall relative abundance within the community matrix. Co-dominants0F included *Sorghastrum nutans* (7.0%), *Carex lurida* (5.9%), *Juncus effusus* (5.7%), *Panicum virgatum* (4.3%), *Eupatorium capillifolium* (3.6%), *Symphyotrichum racemosum* (3.6%), and *Solidago altissima* (3.5%). The Sørensen similarity matrix for the *Lespedeza* dataset showed that community composition was somewhat similar across the invasion gradient (Table 4(a)), with all values above a similarity cutoff of 0.5 [63]. ANOSIM suggested a significant between-group difference based on permutations (*p* = 0.006). From inspection of the ANOSIM boxplots (Figure 5a), nearly all between-group variation was attributable to the A (most invaded) and B (second most invaded) groups, but C (moderately invaded), D (second least invaded), and E (uninvaded) groups were strongly aligned with between-group similarity and were therefore compositionally similar.

*Lonicera*: The *Lonicera* community dataset included 167 species sampled from 50 plots across 10 transects. *Lonicera* comprised 21.9% of the overall relative abundance within the community matrix. Co-dominants included *Liquidambar styraciflua* (5.1%), *Andropogon virginicus* (5.0%), *Rubus pensilvanicus* (4.7%), *Dichanthelium clandestinum* (4.5%), *Juncus effusus* (4.2%), *Parathelypteris noveboracensis* (4.1%), and *Lindera benzoin* (4.1%). As with the *Lespedeza* analysis, the *Lonicera* Sørensen matrix showed marginal compositional similarity across the invasion gradient (Table 4(b)). ANOSIM results demonstrated a significant between-group difference (*p* = 0.001), and boxplots indicated that this difference was due to the invaded groups (A and B), with C, D, and E groups being compositionally similar (Figure 5b).

*Microstegium* (streams): The *Microstegium* stream community matrix included 191 species sampled from 55 plots across 11 transects. *Microstegium* accounted for 30.0% of the overall relative abundance, with co-dominants *Dichanthelium clandestinum* (11.5%), *Solidago altissima* (4.8%), and *Carex lurida* (4.6%). As above, the *Microstegium* community matrix showed marginal similarity in species composition across the invasion gradient based on Sørensen index values (Table 4(c)). ANOSIM results showed significant between-group variation (*p* = 0.001), and boxplots indicated that nearly all between-group variation was due to groups A and B, with the remaining groups showing overlap and compositional similarity (Figure 5c).

### 3.6. Community Properties—Streams

*Lespedeza*: In the *Lespedeza* community matrix, native species richness and FQI peaked at moderate levels of invasion (group C) across the gradient (Table 5). These results accord with species accumulation curves and Rényi diversity profiles, which showed moderately invaded sites (group C) with the highest species richness (Figure 6a), diversity, and evenness (Figure 6d). All results in the *Lespedeza* dataset confirm that the highest levels of *Lespedeza* invasion negatively affect community properties.

*Lonicera*: Like *Lespedeza*, native species richness and FQI were highest at moderate levels of invasion for *Lonicera* (Table 5). Likewise, species accumulation curves showed a clear pattern of species richness values with moderately invaded sites (group C) corresponding to the highest levels of richness across the dataset (Figure 6b). Rényi diversity profiles suggested similar results, although group C diversity values overlapped with group D (second least invaded) and group E (uninvaded) values (Figure 6e). These results also confirm that the highest levels of *Lonicera* invasion negatively affect community properties.

*Microstegium* (streams): As above, the *Microstegium* community matrix showed highest native species richness and FQI values at moderate levels of invasion (group C; Table 5). Species accumulation curves and Rényi profiles for the *Microstegium* dataset accorded with these results, showing the moderately invaded group (C) differentiated as the most species-rich and most diverse along the invasion gradient (Figure 6c,f). As with *Lespedeza* and *Lonicera*, the highest levels of *Microstegium* invasion corresponded to the lowest levels of these community metrics with the exception of group E (uninvaded), which had the lowest species richness profile.

### 3.7. Invasive Species Impact Threshold

For each target invader, a polynomial trendline fitted to the scatterplot of native species richness across the invasion gradient showed that approximately 10% invasion (i.e., 10% relative abundance of the invader) typically coincided with a maximum native richness or the start of the declining limb on the richness curve. These results are shown in Figure 7a–f, with the 10% line depicted on each graph.

## 4. Discussion

This study was focused on sampling changes in the vegetation community across the invasion gradients of different invaders on multiple wetland and stream mitigation sites. Our investigation into floristic quality parameters shed new light on how plant invaders impact ecosystem functions derived from the vegetation community, with potential implications for how invasive plants are managed in these systems.

### 4.1. Impact of Invaders on Species Composition

One of the most interesting results from this study was that the invasion gradient did not reflect the types of changes in species composition that we would have anticipated based on the invasion literature [75,76]. It is important to remember that composition looks at the *identity* of the species present and does not consider numbers of species or their relative abundances, both of which will be addressed below. However, composition has been identified as an important factor in ecosystem function [77] and vegetation development on mitigation sites [29,78].

In the wetland mitigation study, Sørensen similarity coefficients were consistently near or above a rule-of-thumb threshold of 0.5 for this index across all datasets [63], yet we had expected the pairings between the invaded and uninvaded ends of the gradient to be closer to 0. A similarity index close to 0 would have indicated that invasion had changed the composition of species present due to density-dependent effects or habitat modification by the dominant invader, which is not what we found. The index, however, was also not close to 1 in any of the intergroup pairings (i.e., no evidence of high compositional similarity), so the more rigorous computational analysis of ANOSIM was warranted to detect statistical differences that the similarity index by itself might have missed. As the ANOSIM results showed, there was a significant difference between groups, with the difference being attributable to the most invaded plots (A in the case of *Arthraxon* and *Typha*; A and B in the case of *Microstegium*). This result suggests that a “threshold of dominance” needs to be exceeded before *species composition* is affected by the presence of an invader. As Table 3 indicates, that threshold could be high for *Arthraxon* and *Typha* (group A relative abundance = 66.4% and 58.2%, respectively) and reasonably high for *Microstegium* (group B relative abundance = 28.8%). We can conclude, therefore, that invasion does reduce ecosystem functions related to species composition on wetland mitigation sites, but perhaps at a higher level of invasion than previously thought. This conclusion accords with studies documenting community properties in the presence of *Arthraxon* [27,32] and *Typha* [28,39,79].

Irrespective of the dominance threshold concepts noted above, one clear result from our wetland mitigation analysis is that *moderate levels of invasion do not change species composition* on wetland mitigation sites. In all cases, group C (moderately invaded) was compositionally similar to groups D and E (low/no invasion). This suggests that moderate levels of invasion (ca. 5–10%) do not preclude other species from “participating” in the community.

These results were mirrored by the stream study: Sørensen similarity coefficients showed marginal similarity for nearly all inter-group pairings across the invasion gradients in all datasets, and further analysis with the ANOSIM statistic identified that a significant difference between groups was that attributable to the most invaded end of the gradient (groups A and B). Thus, for streams, like wetlands, there appears to be a threshold of dominance beyond which species composition is affected by the presence of an invader, which could be relatively high for the target invaders (group B relative abundance = 26.1%, 38.5%, and 44.1% for *Lespedeza*, *Lonicera*, and *Microstegium*, respectively; Table 5). From these analyses, the conclusions are the same—on stream mitigation sites, the presence of invaders impacts species composition at high levels of invasion, but not at moderate or low levels. Like the wetland results, there was compositional similarity between group C (moderately invaded) and groups D and E (low/no invasion) in the stream datasets. Although the range of group C invader abundance was larger for the stream study (3.2% for *Lespedeza* to 14.0% for *Microstegium*), the average condition still suggests that a 5–10% rule-of-thumb definition for “moderate level of invasion” is reasonable based on both the wetland and stream analyses.

### 4.2. Impact of Invaders on Community Properties

Species richness and diversity are commonly thought of as intrinsic indicators of ecosystem function, in that higher richness and diversity values generally coincide with other important properties such as habitat complexity and ecosystem resiliency [80,81]. Because native species richness is a metric that is generally regarded as important in evaluating wetland mitigation performance [12,71,73], it was a focal point for our research on community properties and invasion. In addition, FQI has been shown to reflect ecosystem function on wetland mitigation sites in Virginia [71].

From the wetland study, our finding that moderate levels of invasion (group C) coincided with maximum native species richness, FQI, and species diversity for all three invaders was unexpected (Table 3 and Figure 4). Although the literature on plant invasion in wetlands is limited with respect to invasion gradients, from the information that is available (e.g., [3,82]), we would have expected a monotonic increase in richness, FQI, and diversity from the invaded to uninvaded ends of the gradient, not a peak in the middle as found. The reasons for the high values of these indicators on the fringes of invasive populations are not clear, but we suspect that localized stress-disturbance dynamics from environmental variation combine to keep more “players in the game” at intermediate levels of invasion. As noted above, disturbance is a factor on nearly all wetland mitigation sites given the nature of the activities that are typically used to modify landforms and augment hydrology regimes [83]. Although difficult to study directly, there are likely localized “disturbance gradients” that coincide with effects from construction or management practices; e.g., staging areas and haul roads can result in increased soil compaction, stormwater discharge points can increase sedimentation and nutrient availability, etc. If these types of localized phenomena were present and able to be diagnosed on our sites, then the arrival and establishment of invaders could have been predicted and even pinpointed based on the literature [11,84]. It is tempting to view the hump-shaped relationship between floristic quality indicators and the center of the invasion gradient as a localized expression of the intermediate disturbance hypothesis, i.e., that species richness and diversity are maximized at intermediate levels of disturbance [85,86], or a variant thereof that includes the interposition of stress and disturbance [62,87].

As with the wetland study, moderate levels of invasion (group C) coincided with maximum native species richness, FQI, and diversity for all three invaders in the stream analyses (Table 5 and Figure 6). The effects of localized disturbance could also be implicated in stream mitigation settings; if so, the open energy cycles [88], variability of cross-sectional and longitudinal gradients [89], and allochthonous influences from watershed inputs [90] would perhaps make construction phase diagnosis of localized disturbance—and attendant invasion risk—more challenging in stream mitigation scenarios vs. wetland sites. Regardless, the stress-disturbance dynamic discussion above is plausible for stream sites, and likely even more relevant given the expected return intervals for disturbance-inducing events like floods [91]. Likewise, it is equally tempting to view pattern and process in community assembly corresponding to the influences of intermediate disturbance, at least at a local scale within the riparian zone of a stream mitigation project [92,93].

Regardless of ultimate cause, it is clear from our results that moderate levels of invasion coincide with high levels of native richness, diversity, and floristic quality in compensatory mitigation. We can conclude, then, that while the presence of invasive species on wetland and stream mitigation sites does affect ecosystem functions related to species richness, diversity, and floristic quality, it only *reduces* these functions at higher levels of invasion. This means that “low threshold” invasive species performance standards, i.e., setting very low tolerances for invasive species performance like 5%, are not advisable based on our results.

### 4.3. Results-Based Invasive Species Performance Standard

The impacts of high levels of invasion on vegetation community functions were anticipated based on the literature and, frankly, common sense; we can see the effects of dominant invaders reflected in their overwhelming density on sites and the resultant diminishment of native species. Collecting data along invasion gradients has allowed us to confirm and enumerate these impacts at the invaded end of the continuum.

What was unexpected in our data was the recurrence of higher levels of these intrinsic floristic quality parameters at moderate levels of invasion. This result was made all the more surprising by the fact that it was consistent across both the wetland and the stream datasets, and for a diverse selection of invaders with respect to habit, life history strategy (e.g., annual grasses, perennial forbs, tall emergent graminoids, woody vines), and tolerance for environmental gradients (e.g., *Typha* and *Lespedeza* living at opposite ends of the moisture continuum).

Programs that have adopted a low threshold like “no greater than 5% cover of invasive species” have done so as a precautionary approach. At face value, the logic of this tactic seems sound; recognizing that a “zero tolerance” stance with respect to plant invaders is likely unattainable [16], a “low tolerance” threshold would provide some latitude for mitigation sites to meet standards while also keeping invaders below a dominance threshold to reduce risk. However, our results suggest that this approach is inherently flawed because it compels mitigation site managers to remediate a condition that is not impacting plant community functions. The most common corrective approach is to use non-selective herbicides to control invasive populations [17], but what our research has shown is that herbicide use (or other methods such as mechanical removal) to control problematic species at moderate levels of invasion will cause indiscriminate mortality of desirable native species at much higher richness, diversity, and floristic quality than previously thought. In addition, continued use of herbicides in long-term management strategies to meet aggressive performance standards has been shown to result in chronic and deleterious effects on environmental conditions such as lower soil nutrient status, decreased dissolved oxygen, acidification, and carbon imbalance, and has in some cases facilitated re-invasion of treated areas [17,18,94].

In light of these considerations, we recommend an invasive species performance standard of 10% relative abundance for invasive plants on both wetland and stream compensatory mitigation sites. Based on our data (Figure 7), a 10% invasive species standard would be a sensible target for ecological performance that strikes a balance between proactive management and indiscriminate loss of desirable species and ecosystem function.

### 4.4. Monitoring Plant Invasion on Mitigation Sites—Recommendations

It is important to point out that we are not advocating for invasive plants to be ignored until a 10% invasion threshold is crossed on mitigation sites. In application, a 10% performance standard should be monitored by calculating the relative abundance of invasive species from vegetation monitoring data (e.g., plot-based data or equivalent), as long as the monitoring data have been collected using methods that conform to assumptions of ecological sampling theory for which sample adequacy has been demonstrated [95]. This means that on most mitigation sites, invasive species will be tracked by community type or planting zone rather than by the site as a whole. On sites where more than one invasive species is present, relative abundance should be calculated as a cumulative value for the performance standard (i.e., sum of relative abundance values for all invasive species present).

The benefit of assessing invasion using a randomized sample dataset that has been subjected to a sample adequacy test is that it provides an unbiased estimate of invader abundance. This type of surveillance is advisable because it discourages the habit of monitoring stationary plots from year to year and, as a consequence, only treating invasive species in localized proximity to monitored plots. However, a monitoring program that includes both random samples *and annual mapping of invader populations* would be the best approach to reducing the risk of invasion on mitigation sites. Combining relative invader abundance with mapping of invasive species to determine the areal extent of localized “invasion hot spots” on a site would be a judicious approach that, in our opinion, would not be excessive in comparison with typical vegetation monitoring requirements. Most compulsory mitigation monitoring is completed using plot-based sampling techniques, the results of which are summarized in data tables that can be manipulated to calculate relative abundance. Likewise, most monitoring requirements for mitigation projects stipulate updated mapping of site resources per monitoring year (e.g., jurisdictional wetland limits, vegetation communities, groundwater well and plot locations, etc.), and invasive species populations are frequently included in that effort.

## 5. Conclusions

On compensatory mitigation sites, the presence of plant invaders impacts vegetation community functions at high but not moderate levels of invasion. This finding holds for both wetland and stream mitigation sites and for a range of target invaders with different growth forms, ecological tolerances, and life history strategies. These results suggest that compulsory invasive species management triggered by low invasive species ecological performance standards (e.g., 5% or lower based on our data) would result in a loss of desirable native species that is not compensated for by the reduction in the target invader and could cause further degradation due to the cumulative effects of chemical herbicide use in natural systems. Based on our data, an ecological performance standard of 10% relative abundance of the invader(s) would be an appropriate target that would balance the equally important objectives of invasive species management and maintenance of vegetation community functions. We also recommend that a proactive invasive species mapping program be combined with annual quantitative vegetation monitoring.

## Figures and Tables

**Figure 1 biology-13-00275-f001:**
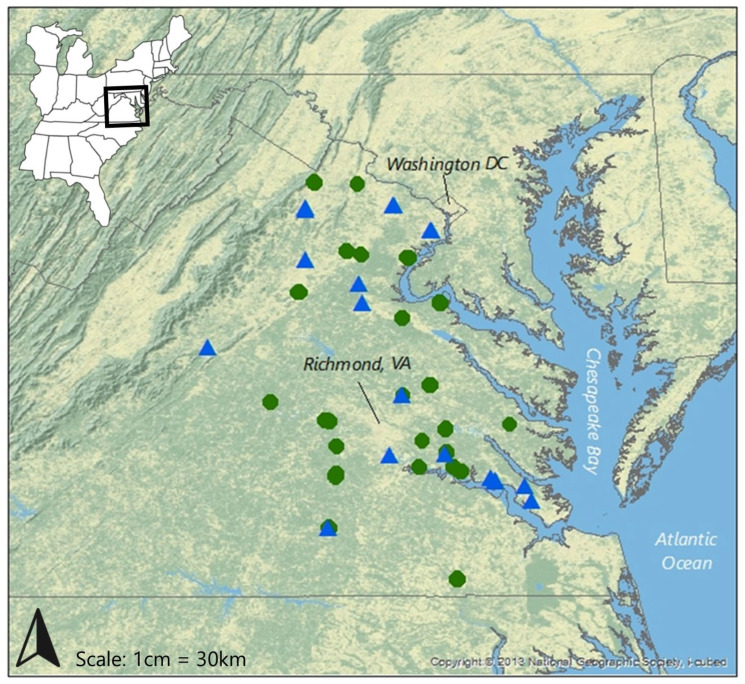
Study site locations. Green circles indicate wetland mitigation sites and blue triangles are stream mitigation sites.

**Figure 2 biology-13-00275-f002:**
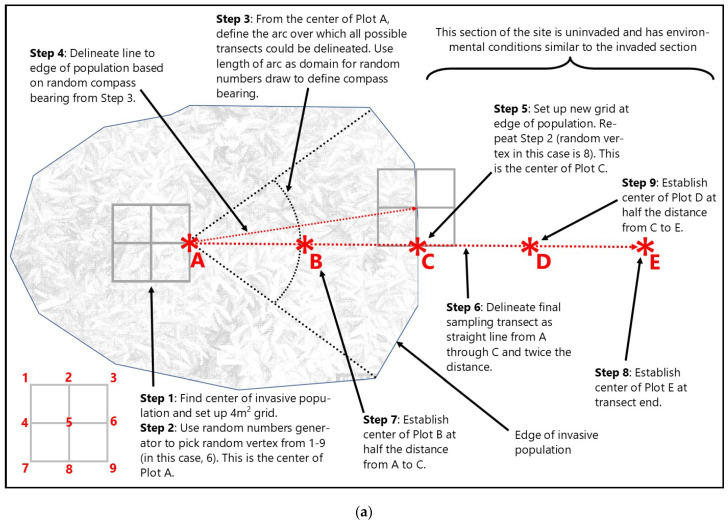
General layout of wetland (**a**) and stream (**b**) study design and transect/plot configuration. Transect orientation, plot randomization, and final plot position (red asterisks) explained in text (Section 2.3.1 and Section 2.3.2).

**Figure 3 biology-13-00275-f003:**
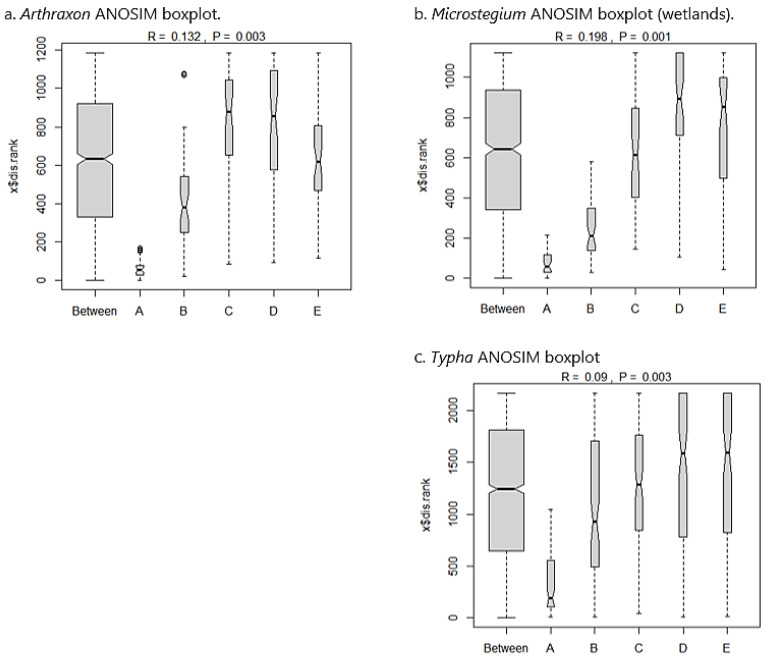
ANOSIM boxplots for the wetland datasets showing distribution of compositional similarity among groups across the invasion gradient from most invaded (A) to uninvaded (E). For each dataset, differences in species composition from the ANOSIM statistic are attributed to groups A and B (*Microstegium*) or group A only (*Arthraxon*, *Typha*), with moderately invaded (C) sites showing compositional affinity to the uninvaded end of the gradient and strong overlap with between-group similarity. Boxplot width is proportional to the number of observations per group (“Between” being the largest as it includes all plots across groups). Notch corresponds to group median, and whiskers show group distribution (outliers greater than 1.5 times the interquartile range are plotted as points).

**Figure 4 biology-13-00275-f004:**
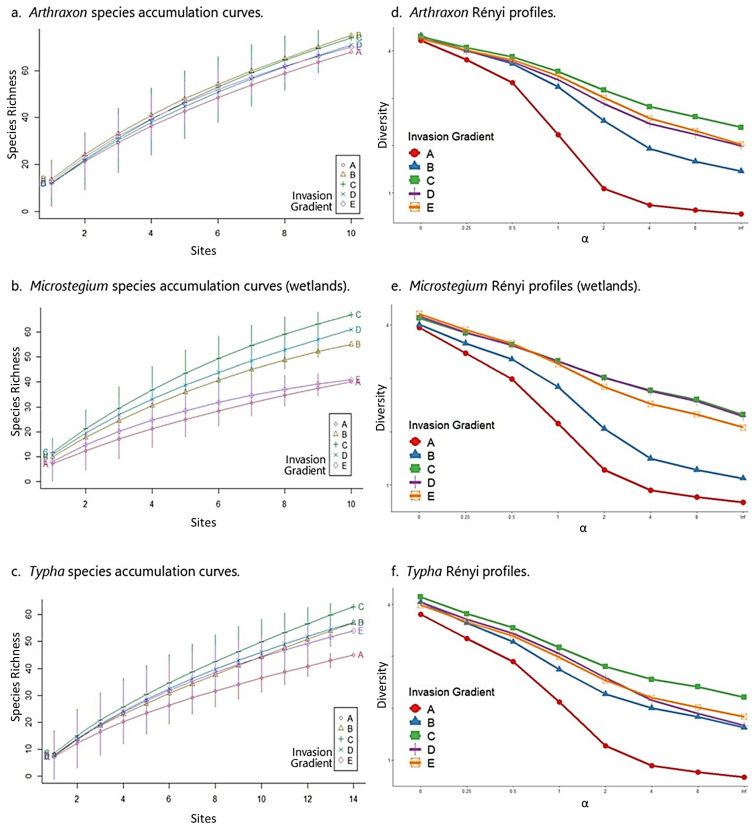
Species accumulation curves and Rényi profiles for the wetland datasets. In each graph, the invasion gradient is represented by the different curves from A (most invaded) to E (uninvaded). The highest curves on the species accumulation and Rényi graphs represent the highest species richness and diversity values, respectively. The x-axis on the Rényi graphs is a unitless diversity ordering scale referred to as alpha (α). It represents species richness (α = 0, left hand side), Shannon diversity index (α = 1, center), Simpson diversity index (α = 2, center), and species evenness (α = inf., right hand side), all of which represent transformed values of those original metrics to make them proportional and thus representable on one graph.

**Figure 5 biology-13-00275-f005:**
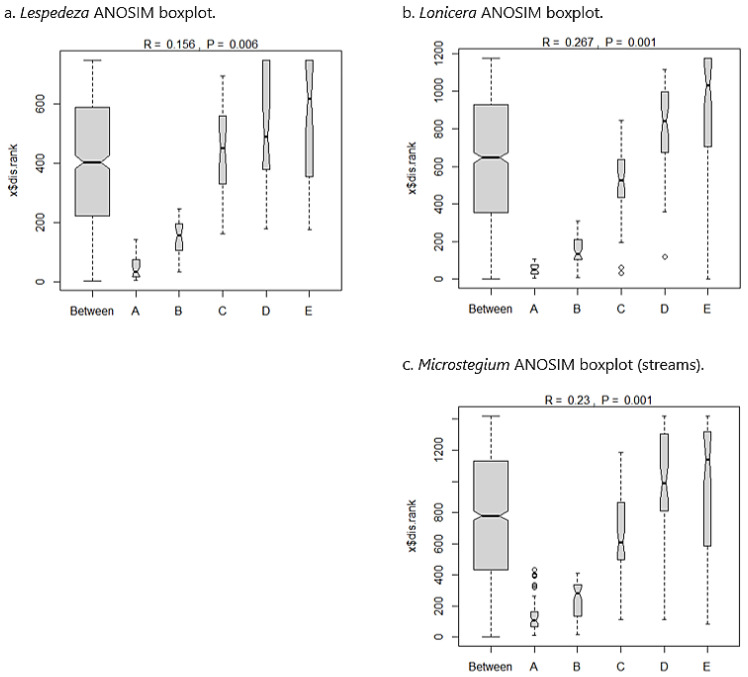
ANOSIM boxplots for the stream datasets showing distribution of compositional similarity among groups across the invasion gradient from most invaded (A) to uninvaded (E). For each dataset, differences in species composition from the ANOSIM statistic are attributed to groups A and B, with moderately invaded (C) sites showing compositional affinity to the uninvaded end of the gradient and a strong overlap with between-group similarity. See Figure 3 caption for additional information on boxplot interpretation.

**Figure 6 biology-13-00275-f006:**
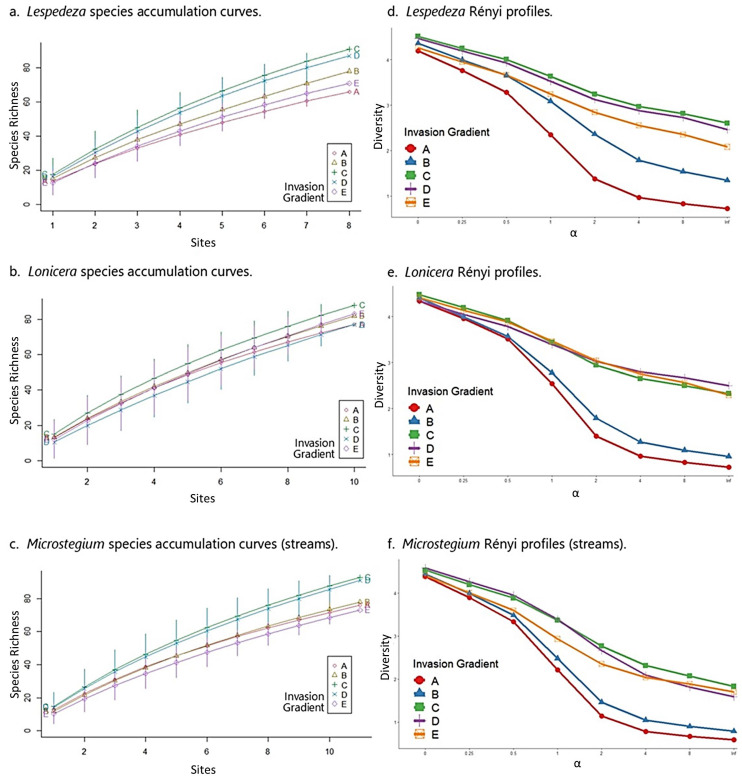
Species accumulation curves and Rényi profiles for the stream datasets. See Figure 4 caption for comments on interpretation.

**Figure 7 biology-13-00275-f007:**
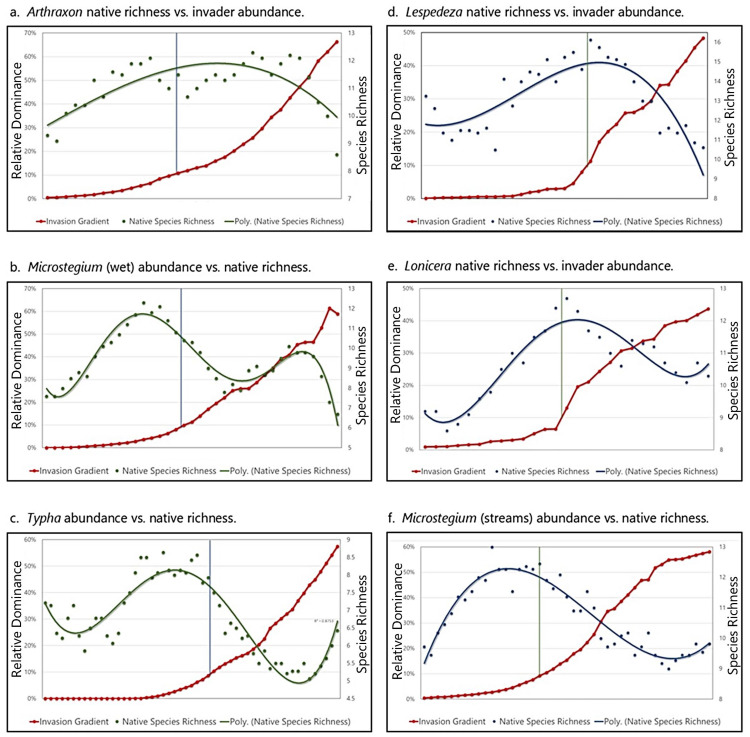
X–Y scatterplot of mean native species richness and invasive species abundance for all taxa in both the wetland and stream studies. On each graph, the vertical “10% threshold” line is projected from the invasion gradient (red line) upward and intersecting with the native richness polynomial trendline for wetlands (green) and streams (blue). In all cases, the 10% line coincides with the peak or the start of the receding limb for the “hump” in the native richness curve.

**Table 1 biology-13-00275-t001:** Target plant invaders studied on both wetland and stream mitigation sites. Dur. = “duration”; A = annual; P = perennial.

Group	Scientific Name	Family	Dur.	Habit	Origin	Comments	Refs.
Wetland mitiga-tion	*Arthraxon hispidus* (Thunb.) Makino	Poaceae	A	Graminoid	East Asia	Little attention in the literature. Moderately invasive throughout Mid-Atlantic region; problematic on mitigation sites.	[30,31,32]
	*Microstegium vimineum* (Trin.) A. Camus	Poaceae	A	Graminoid	Asia	Tolerant of shading and temporary flooding. Prolific seeder. Forms persistent seed banks. Inhabits variety of wetland and upland habitats. Alters community structure and reduces native plant diversity.	[31,33,34,35,36,37,38]
	*Typha* spp. L.	Typhaceae	P	Forb	U.S.	Two cattail species (*Typha latifolia* L. and *Typha angustifolia* L.) and their hybrid (*Typha* x *glauca* Godron). Native to U.S. (*T. angustifolia* putatively introduced from Europe) but regulated as invasive on wetland mitigation sites. Tolerant of prolonged inundation. Impacts on wetland communities have been questioned.	[3,28,39,40,41,42,43,44]
Stream mitigation	*Lespedeza cuneata* (Dum.-Cours.) G. Don	Fabaceae	P	Forb	East Asia	Occupies well-drained soils. A nitrogen fixer, its extensive taproot allows survival in drought conditions and a wide range of soil pH. Modifies habitat to facilitate invasion. Herbivory-resistant with allelopathic properties. Modifies nutrient pools by rapid uptake/slow release via slower decomposition (through concentrated tannins and phenolic compounds).	[30,31,45,46,47,48,49,50,51]
	*Lonicera japonica* Thunb.	Caprifoliaceae	P	Vine	East Asia	Dispersed by birds but expansion generally occurs vegetatively. Due to high transpiration rates, does not tolerate prolonged drought and therefore tends to prefer mesic habitats, making riparian zones, streambanks, and floodplains susceptible to invasion. Somewhat shade tolerant, but prefers canopy openings to promote localized dominance.	[30,31,52,53,54,55,56,57,58,59]
	*Microstegium vimineum* (Trin.) A. Camus	Poaceae	A	Graminoid	Asia	See comments above. Factors contributing to invasion potential are perhaps more important in streams, floodplains, and riparian zones due to use of flowing water as a dispersal mechanism.	[26,33,37,60]

**Table 2 biology-13-00275-t002:** Sørensen similarity matrices for wetland datasets across the invasion gradient from A (most invaded) to E (uninvaded).

a. *Arthraxon*		B	C	D	E
	A	0.57	0.59	0.67	0.44
	B		0.61	0.60	0.57
	C			0.56	0.46
	D				0.50
b. *Microstegium*		B	C	D	E
	A	0.48	0.56	0.52	0.58
	B		0.48	0.64	0.50
	C			0.52	0.51
	D				0.56
c. *Typha*		B	C	D	E
	A	0.65	0.49	0.64	0.57
	B		0.51	0.57	0.54
	C			0.56	0.57
	D				0.70

**Table 3 biology-13-00275-t003:** Mean native species richness, floristic quality index (FQI), and mean relative abundance of invader across invasion gradient from A (most invaded) to E (uninvaded) on wetland mitigation sites. Moderate invasion (C, red typeface) corresponds to the highest values of native species richness and FQI in the datasets of all three invaders.

Mean Native Species Richness					
Invasion Gradient:	A	B	C	D	E
*Arthraxon*	8.6	11.9	** 12.1 **	9.5	12.4
*Microstegium*	5.7	8.3	** 9.8 **	8.6	8.4
*Typha*	6.1	5.4	** 9.0 **	5.9	7.2
Floristic Quality Index (FQI)					
Invasion Gradient:	A	B	C	D	E
*Arthraxon*	10.4	12.7	** 12.8 **	11.3	12.2
*Microstegium*	9.4	11.0	** 12.3 **	11.8	11.6
*Typha*	8.5	7.6	** 10.1 **	9.3	9.1
Mean Native Species Richness					
Invasion Gradient:	A	B	C	D	E
*Arthraxon*	66.4	23.2	** 6.6 **	0.7	0.0
*Microstegium*	58.9	28.8	** 8.1 **	0.9	0.0
*Typha*	58.2	26.6	** 5.1 **	0.0	0.0

**Table 4 biology-13-00275-t004:** Sørensen similarity matrices for stream datasets across the invasion gradient from A (most invaded) to E (uninvaded).

a. *Lespedeza*		B	C	D	E	
	A	0.61	0.57	0.58	0.69	
	B		0.70	0.67	0.66	
	C			0.67	0.62	
	D				0.66	
b. *Lonicera*		B	C	D	E	
	A	0.58	0.56	0.48	0.46	
	B		0.64	0.52	0.53	
	C			0.63	0.55	
	D				0.65	
c. *Microstegium*		B	C	D	E	
	A	0.56	0.51	0.49	0.53	
	B		0.54	0.55	0.50	
	C			0.62	0.55	
	D				0.56	

**Table 5 biology-13-00275-t005:** Mean native species richness, floristic quality index (FQI), and mean relative abundance of invader across invasion gradient from A (most invaded) to E (uninvaded) on stream mitigation sites. Moderate invasion (C, red typeface) corresponds to the highest values of native species richness and FQI in the datasets of all three invaders.

Mean Native Species Richness					
Invasion Gradient:	A	B	C	D	E
*Lespedeza*	11.4	13.3	** 15.1 **	14.4	10.6
*Lonicera*	10.3	10.7	** 12.4 **	8.6	10.1
*Microstegium*	9.8	9.5	** 12.1 **	11.9	9.0
Floristic Quality Index (FQI)					
Invasion Gradient:	A	B	C	D	E
*Lespedeza*	11.1	12.8	** 14.0 **	11.5	12.6
*Lonicera*	10.7	12.4	** 13.5 **	10.2	11.3
*Microstegium*	11.1	11.1	** 12.9 **	12.2	10.7
Mean Native Species Richness					
Invasion Gradient:	A	B	C	D	E
*Lespedeza*	48.4	26.1	** 3.2 **	0.6	0.2
*Lonicera*	48.6	38.5	** 6.5 **	1.1	0.4
*Microstegium*	56.9	44.1	** 14.0 **	2.6	0.3

## Data Availability

The datasets generated during and/or analyzed during the current study are available from the corresponding author upon request.

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
