# Peer review of "Impacts of Invasive Plants on Native Vegetation Communities in Wetland and Stream Mitigation"

_biology, 2024, doi:10.3390/biology13040275_

Round 1

Reviewer 1 Report

Comments and Suggestions for Authors

In the study entitled “Impacts of Invasive Plants on Native Vegetation Communities in Wetland and Stream Mitigation« submitted to the journal »Biology« (Manuscript ID: biology-2937556) the authors address the impact of selected invasive plant species on native plant communities in terms of plant species composition, richness, floristic quality and diversity.

Notably, the study reveals that low and moderate levels of invasion do not adversely affect native plant communities in wetland and stream mitigation sites. This research is compelling, relevant, and timely in the context of globalization and the resulting bioinvasions, significantly enhancing our understanding of the effects of invasive species on native plant communities. Furthermore, it offers valuable insights for effective management in mitigation settings and provides recommendations for future conservation efforts.

The paper is well-organised and very well-written. The authors adequately introduce the topic and demonstrate on examples the practical need for the study, select appropriate sampling and statistical methods and represent results systematically. The discussion supports the findings of the study.

However, I have included some minor comments in the attached file for consideration.

Author Response

The suggested revisions from Reviewer 1 were very helpful and have improved the manuscript.  Below are point-by-point responses:

1- What are the thresholds in Virginia? Text was added to Section 1.1 to include the 5% Virginia threshold.

2- l. 134 Are not all invasive species considered in the study(s) non-native? The word "non-native" was removed from the text to limit confusion. However, suffice it to mention that there are important invasive plants in the eastern US that are considered traditionally "native" to the region (e.g., Phalaris and Typha). 

3- What criteria were used to select the target invasive species?  Narrative on the species selection process was added to Section 2.1. 

4- Can you provide more details about suitability criteria?  Narrative on site suitability criteria was added to Section 2.2 (first paragraph).

5-  Can the table be rotated clock-wise?  This change was made as suggested.

6- Include a scale and compass (north arrow). 
I suggest also to include a scaled-down area to help orient global readers.  North arrow, scale, and inset map added to Figure 1.

7- Figure 5, 6, 7: Axis labels are too small...  This comment was actually posted at Figure 4, and we agree that Figures 4, 6, and 7 had this problem.  All figures have been revised to increase the axis label font sizes.

8- The reviewer underlined the words "sericea lespedeza" at line 735. It is not clear what the recommended edit is here - sericea lespedeza is a common name and therefore would neither be italicized or capitalized. Also, the title of the paper cited at line 735 was reproduced verbatim from the original paper. 

Reviewer 2 Report

Comments and Suggestions for Authors

In my opinions this MS deserves to be published on Biology after minor formal revision.

The authors have not followed the istructions in preparing the manuscript

https://www.mdpi.com/journal/biology/instructions

In addition in the text and in the table the auhtors of the scientific names have to be added.

Author Response

We appreciate the recommendations from Reviewer 2 and have made edits accordingly.  We respectfully submit the following responses to each comment:

1- add the authors of the names. 

All scientific names in the appendices have been updated with authorities.

2- Add if the plant is Casual, Naturalized or Invasive. 

We changed the far right column header to "Status", which is defined in the table footnote as "Status = "Native", "Non-native", or "Invasive" in accordance with Heffernan et al. (2014) and Weakley et al. (2020)."  We interpreted the reviewer's comment about "Casual" introductions to mean adventive species (i.e., introductions that have not naturalized), however, there were no adventive species in either data set. 

Reviewer 3 Report

Comments and Suggestions for Authors

General comment

It is a detailed and fine work with accurate data important in the specific context of vegetation restoration management in a region of the United States. I think that in order to show the importance and applicability of the results to different areas and countries, broadening the interest for readers, some more information and context of the sites, vegetation and invasion processes will help.

Section Materials and Methods

Lines 158-168: Can the authors give some more information about the sites in the two areas studied (piedmont and coastal plain)? This will help readers out of the US to better understand the study. Are the different sites more or less similar in history and perturbation events or drivers? What do authors mean by ‘construction’ of the site? Is it when the mitigation site was formally established? Does it imply any intervention (and which ones) on the vegetation at the site at the time of construction or from that date on?

Line 252. Maybe authors can start the Results section giving a overall view of the results, in terms of the invaders and invasion. This can be supported by a table with the general data for the studied species, gathering the information from the first phrases in the paragraphs of species (lines 257-262, 284-288, 293-296). The same is suggested for section 3.5, maybe in the same table.

Line 331: Add ‘Floristic Quality Index’ also in the Table’s description.

Line 337: Figure 4. Maybe authors can explore to represent the Invasion Gradient along the X axis, so the reader would better visualize the effect of the gradient on richness and diversity. If the numerous curves became less clear, perhaps a single average value can be represented in the graph.

Authors may deepen the Discussion giving a bit more information about the management practices. Is there evidence of how herbicide application works in restoration for the study cases or in wetlands/streams of that area? Can they suggest how a change in threshold management would influence vegetation restoration in the medium and long term?

Line 553: The ‘time’ dimension is not clear in the manuscript. As it its important, I suggest the authors give some idea about (briefly describe) how the invasion occurs over time for the study species, probably in sections 2.1 or 2.2.

Also in Discussion. As far as I understand, the invasion at a given site occurs in a rather restricted spot, since there is a gradient from the highest invaded ‘center’ of invasion to the uninvaded end of the gradient. How does the herbicide or management practice is applied? Do they apply herbicide only to the invaded area, or to the whole restoration site?

It is also important to have an idea of the target restoration (future state/condition) in terms of the vegetation expected to be attained after a successful process of restoration. Does it include trees, shrubs, vines, a prairie? Maybe this information will help the readers to understand why a management with herbicides is chosen at present.

Author Response

We are very appreciative of the reviewer's comments.  We believe the changes made have greatly improved the manuscript accordingly.  Below are point-by-point responses.

Lines 158-168: Can the authors give some more information about the sites in the two areas studied (piedmont and coastal plain)? This will help readers out of the US to better understand the study. Are the different sites more or less similar in history and perturbation events or drivers? What do authors mean by ‘construction’ of the site? Is it when the mitigation site was formally established? Does it imply any intervention (and which ones) on the vegetation at the site at the time of construction or from that date on?

Clarifying text was added to Section 2.2.

Line 252. Maybe authors can start the Results section giving a overall view of the results, in terms of the invaders and invasion. This can be supported by a table with the general data for the studied species, gathering the information from the first phrases in the paragraphs of species (lines 257-262, 284-288, 293-296). The same is suggested for section 3.5, maybe in the same table.

We appreciate this suggestion; however, it is our opinion that the most relevant information is already summarized in Tables 2-5. We believe that re-stating the results from those sections of the narrative in a table would unnecessarily lengthen the manuscript.

Line 331: Add ‘Floristic Quality Index’ also in the Table’s description.

Edit made to Table 3 and Table 5.

Line 337: Figure 4. Maybe authors can explore to represent the Invasion Gradient along the X axis, so the reader would better visualize the effect of the gradient on richness and diversity. If the numerous curves became less clear, perhaps a single average value can be represented in the graph.

This is in reference to the “invasion gradient”, which is represented in the figures by the position of each plot along the sampling transects (A=most invaded,…B…,…C…,…D…, E=uninvaded). Therefore, each individual line on the graph can be interpreted as a representing a level of invasion. It would not be appropriate to re-orient the species accumulation curves as the purpose of species accumulation curves is to demonstrate how the number of species increases as additional sites (plots) are added from left to right. The same can be said for the Rényi profiles – the purpose is to visualize the full spectrum of richness, evenness, and diversity on one graph.  Reconfiguring the Rényi figures would diminish their utility.  We are hoping that the additional detail provided in the figure caption should explain the intent of these figures to the reader and make their current configuration clear.

Authors may deepen the Discussion giving a bit more information about the management practices. Is there evidence of how herbicide application works in restoration for the study cases or in wetlands/streams of that area? Can they suggest how a change in threshold management would influence vegetation restoration in the medium and long term?

We believe that this topic is already addressed in Section 4.3.  If the reviewer disagrees, we are happy to consider adding narrative.

Line 553: The ‘time’ dimension is not clear in the manuscript. As it its important, I suggest the authors give some idea about (briefly describe) how the invasion occurs over time for the study species, probably in sections 2.1 or 2.2.

Removed “in space and time” from this sentence.  We appreciate the reviewer’s request here, and agree that a summary of these characteristics would be beneficial if it could be summarized.  However, the invasion sequence is too highly variable from species to species, from site to site, and even within an individual species to adequately characterize in a brief synopsis.  What we understand from our research and the growing literature on this topic is that the common denominator is disturbance, and nearly all invaders in our study are quick to take advantage.  We believe this is already adequately addressed in the introduction and again in Section 4.2.

Also in Discussion. As far as I understand, the invasion at a given site occurs in a rather restricted spot, since there is a gradient from the highest invaded ‘center’ of invasion to the uninvaded end of the gradient. How does the herbicide or management practice is applied? Do they apply herbicide only to the invaded area, or to the whole restoration site?

This is not always the case.  Many invasions occur at multiple points throughout the site, particularly on the stream sites which are more “open” ecosystems (open energy cycles, and exposed to waterborne propagule dispersal via uni-directional flooding).  Our research approach focused on the “restricted spot” idea mentioned by the reviewer so that we could sample across the continuum from invaded to uninvaded in a deliberate way.  The alternative would have been to sample random plots throughout a site, which would have been much less informative for the objectives of the study.  In answer to the reviewer’s question about management, typically contractors do spot treatments with herbicides when populations are smaller.  This approach is generally done with a backpack sprayer unit and can be reasonably accurate and targeted.  However, when the invaders are pervasive and cover larger areas, contractors usually use a broadcast method (typically a boom-mounted sprayer on the back of an all-terrain vehicle).  This technique is non-selective and will kill most plants within the spray profile.

It is also important to have an idea of the target restoration (future state/condition) in terms of the vegetation expected to be attained after a successful process of restoration. Does it include trees, shrubs, vines, a prairie? Maybe this information will help the readers to understand why a management with herbicides is chosen at present.

Additional text was added to Section 2.2.